# Physical Fitness Profiling of Youth Basketball Players by Developmental Stage: A Case Study

**DOI:** 10.3390/jfmk10040382

**Published:** 2025-10-02

**Authors:** Olga Calle, David Mancha-Triguero, Eduardo Recio, Sergio J. Ibáñez

**Affiliations:** 1Grupo de Optimización del Entrenamiento y Rendimiento Deportivo (GOERD), Facultad de Ciencias del Deporte, Universidad de Extremadura, 10002 Cáceres, Spain; olcallem@unex.es (O.C.); eduardo.recio.rol@gmail.com (E.R.); sibanez@unex.es (S.J.I.); 2University CEU Fernando III, CEU Universities, 41930 Sevilla, Spain; 3Faculty of Health Sciences, Universidad Isabel I, 09003 Burgos, Spain

**Keywords:** youth competitive basketball, physiological fitness profiling, SBAFIT battery protocol, athletic performance evaluation, developmental age categories, biological maturation

## Abstract

**Background**: Basketball is characterized as a high-intensity, intermittent sport that places considerable demands on the cardiorespiratory, neuromuscular, and mechanical systems. These physiological requirements are modulated by contextual variables and the athlete’s stage of biological maturation, both of which significantly influence physical fitness outcomes. Consequently, it is imperative to employ age- and development-specific assessment protocols. **Objectives**: This study aimed to evaluate the differences in physical fitness across competitive categories and to explore the interrelationships among the various physical assessment tests. Twenty-four male players (U14 = 12; U16 = 12) participated in this research. **Methods**: Athletes were monitored using WIMUPRO inertial measurement units and completed the SBAFIT test battery to evaluate physical fitness parameters. Statistical analyses included both inferential and correlational approaches, with effect sizes calculated for all relevant variables. The independent variable was the competitive age category of the players. **Results**: The results indicated notable differences in physical performance between developmental groups, primarily attributed to biological maturation. Significant disparities were observed in measures of aerobic capacity, linear speed, agility, and centripetal force. **Conclusions**: The comparative nature of this study across developmental categories offers novel insights and practical implications for talent development and training optimization.

## 1. Introduction

Basketball is a high-intensity intermittent sport characterized by explosive actions such as sprints, jumps, and changes of direction (CODS). These actions alternate with phases of submaximal activity [1]. In this line, Torres-Unda et al. [2] described basketball as a hybrid sport, as athletes alternate between periods in which they obtain energy primarily through the aerobic pathway and other periods in which they obtain energy primarily through the anaerobic pathway. As a result, players are subjected to significant cardiorespiratory, neuromuscular, and mechanical demands, which in turn influence training planning, recovery processes, and injury prevention strategies. Furthermore, the magnitude of these demands is modulated by contextual factors (season phase, specific playing position, or competitive level), highlighting the necessity of precisely characterizing the physiological profile of the sport. Success in basketball depends on various factors, among which morphological attributes [3], physical and technical abilities [4], and tactical actions [5] are particularly prominent.

In line with the considerations, numerous studies have described the physical fitness (PF) of basketball players across different ages, sexes, and competitive levels [6,7,8]. The findings indicate differential profiles in strength, speed, jump power, and intermittent capacity according to sex and biological maturity, with marked improvements as chronological and sports age advance [9,10]. Moreover, longitudinal monitoring within the same season reveals significant fluctuations in aerobic, anaerobic, and external load variables [11], supporting the need for regular assessments to adjust training loads and individual goals [12].

Guimaraes et al. [13] demonstrated that biological maturation status is the primary predictor of physical performance in basketball players aged from 11 to 14 years, while training experience is the most influential factor for technical skills. A recent systematic review has shown that young athletes with early maturation present superior values in PF variables such as vertical jump, handgrip strength, and speed tests, as well as higher body mass, height, and body fat percentage compared to their late-maturing peers. This emphasizes the need for and importance of interpreting physical performance in light of individual maturation status [14]. Similarly, coordination abilities have been shown to play a central role in basketball development, with strong links between general and sport-specific coordination across youth stages [15]. Likewise, systematic monitoring of PF enables the early identification of potential deficiencies or stagnation in players’ physical development, facilitating preventive or corrective interventions before these issues translate into decreased performance or increased injury risk [16,17].

Moreover, a positive relationship between various components of PF and performance indicators during matches (efficiency rating, minutes played, or relative load) has been consistently demonstrated [18,19,20]. Recent studies show, for instance, that players with higher relative power values and superior performances in agility tests are capable of sustaining higher external workloads without impairing their technical performance metrics, thereby evidencing a clear link between PF and competitive effectiveness [21]. These findings reinforce the notion that PF should be regarded as a strategic prerequisite within evidence-based training processes.

The primary objective of PF evaluation is to monitor, control, and validate the physical conditioning of basketball players, enabling the assessment, understanding, and comparison of an athlete’s physical status at different points throughout the season. This information allows coaches to appropriately adjust training methods and resources, thereby contributing to enhanced athletic performance [22]. Nevertheless, PF evaluation often still relies on general laboratory or field tests originally developed for other sports. Although such protocols are useful for inter-sport comparisons, they offer limited relevance to basketball, a sport characterized by highly specific movement patterns, technical actions, and braking demands [23]. The principle of training specificity thus necessitates the adoption of ad hoc testing protocols designed to replicate the mechanical, metabolic, and coordinative stimuli inherent to the game.

Change of Direction Speed (CODS) has been identified as a crucial determinant of physical performance in basketball, with a wide range of tests developed to assess it. Sugiyama et al. [24] classified 48 types of CODS tests into three main categories (defensive, 180° turn, and cutting maneuvers) and emphasized that each type reflects distinct motor demands specific to gameplay, highlighting the need for selecting tests aligned with the most representative technical movements of the sport. In accordance with the classification proposed by Sugiyama et al., the test described should be interpreted as a measure of change of direction (COD) rather than true agility, as it involves pre-planned directional changes without decision-making in response to an external stimulus. In response to this requirement, test batteries such as SBAFIT were developed, incorporating ten field-based tests (aerobic, anaerobic, maximal and reactive strength, linear/specific speed, generic/specific agility, and centripetal force) to provide a comprehensive diagnostic profile of the player [25]. Subsequently, complementary approaches have been proposed, such as the multi-location external load battery, which integrates inertial device analysis to simultaneously quantify neuromuscular load across different joints [26].

Several studies have explored the PF of players during their developmental stages. Erčulj, Blas and Bračič [1] analyzed elite U’15 European female players through generic tests that included ball-handling elements, establishing female-specific reference values regardless of playing position. Calleja-González et al. [27] conducted a two-year longitudinal study with U’16 players to determine whether a systematic training process would yield improvements in neuromuscular and aerobic power among young athletes. Similarly, Mancha-Triguero, Martin-Encinas, and Ibáñez [12] examined the seasonal evolution of aerobic/anaerobic fitness, jump power, and agility using the SBAFIT battery in U’18 female players. Additionally, studies comparing a lactic anaerobic test with two agility tests in U’14 and U’16 female players identified significant relationships among the analyzed variables, recommending the use of shorter tests to measure the same conditional factor [28].

There is a paucity of scientific evidence regarding the evaluation of PF in youth basketball players, despite its growing relevance in contemporary sports science. Most research efforts have focused on assessing the physical condition of players competing in higher-level categories. Although notable advances have been made, the majority of these validations have been conducted on senior or semi-professional populations, while young players (whose maturational development introduces additional variables) remain underrepresented in the literature [29]. The scarcity of specific information pertaining to youth categories constrains the design of evidence-based talent identification programs, load monitoring strategies, and injury prevention protocols. Accordingly, the objectives of the present study were threefold: (i) to describe the physical performance of two youth teams using a basketball-specific test battery, (ii) to compare the performance outcomes of the two teams according to their age groups, and (iii) to identify the relationships among the variables assessed within the test battery.

## 2. Materials and Methods

### 2.1. Design

This study constitutes an empirical, quantitative, ex post facto, and cross-sectional investigation [30], whose primary objective was to evaluate the PF levels of basketball players belonging to two age categories (U’14 and U’16) at the conclusion of the season, as well as to compare the differences observed between these two groups.

### 2.2. Participants

Male players from the U’14 (*n* = 12; 168.75 ± 9.13 cm; 13.38 ± 0.48 years) and U’16 (*n* = 12; 185.25 ± 8.55 cm; 15.33 ± 0.47 years) teams of the same club participated in this study. It should be noted that the two groups were not expected to be equivalent at baseline, since age category (U’14 vs. U’16) was the independent variable under analysis; the objective of this study was precisely to examine the physical fitness differences attributable to developmental stage. A convenience sample was used, comprising all available players from each squad; therefore, no a priori sample size calculation was performed. Both teams were selected due to their status as regional benchmarks within their respective categories and their affiliation with a men’s basketball club competing at the highest federative level nationally.

### 2.3. Sample

The sample consisted of the individual results obtained through the SBAFIT test battery, administered across two separate sessions. To collect the data, players were equipped with inertial measurement units (IMUs), specifically the WIMU PRO™ devices (RealTrack Systems, Almería, Spain). Additionally, photocell timing gates (ChronoJump, Barcelona, Spain) were used to enhance the precision of the temporal measurements. The data sample varied depending on the nature of the variables analyzed; in some cases, a single value was recorded, whereas in others, the data depended on the sampling frequency of the IMUs.

### 2.4. Variables

The independent variable of this study was the sport category (U’14 and U’16). The dependent variables were the PF capacities assessed through the SBAFIT test battery [25]. The specific dependent variables selected for the analysis in this study for each test are presented in Table 1.

### 2.5. Instruments

The validated SBAFIT battery of basketball-specific tests [25] was employed to conduct this study. To ensure transparency and reproducibility, a brief description of the SBAFIT battery is provided. The SBAFIT is a validated, basketball-specific field-based test battery composed of ten assessments that collectively measure aerobic capacity (SIG/AER test), anaerobic lactic capacity (SIG/ANA test), maximal and fatigue tolerance of lower-limb strength (Abalakov jump and Multijump tests), general and specific sprint speed (with and without ball dribbling), generic and basketball-specific agility (T-Test, with and without ball dribbling), and centripetal force (Arc Test, with and without ball dribbling). Each test replicates the technicaltactical demands of basketball by integrating sport-specific movements such as dribbling, rebounding, and defensive shuffles, thereby enhancing ecological validity compared with generic protocols. A graphical overview and detailed protocols of the SBAFIT are available in Mancha-Triguero, García-Rubio, and Ibáñez [25].

The Agility Test with and without the ball used in the SBAFIT is classified as a pre-planned Change-of-Direction Speed (CODS) test; the course is fully predetermined and, therefore, does not involve unpredictable external stimuli or the perceptual-cognitive components characteristic of reactive agility.

To record the kinematic and neuromuscular variables, inertial measurement devices (WIMU Pro) were utilized. Additionally, body composition was assessed using the TANITA MC-780MA analyzer (TANITA, Tokyo, Japan), which is based on bioelectrical impedance analysis (BIA).

The processing and downloading of session data were carried out using the SPro software, version 989 (WIMU Pro). Subsequently, the data were recorded in Microsoft Excel (Microsoft Corporation, Redmond, Washington, USA). Finally, all statistical analyses were performed using the Statistical Package for the Social Sciences (SPSS), version 25 (IBM Corp. 2012. IBM SPSS Statistics for Windows, NY: IBM Corp., Armonk, NY, USA).

### 2.6. Procedure

During the initial phase of this study, players, coaches, and strength and conditioning staff were informed about the objectives, risks, and benefits associated with their participation. This study was approved by the Bioethics Committee of the University of Extremadura (87/2023), in accordance with the Declaration of Helsinki (2013), and informed consent was obtained from each participant.

The first step of the experimental phase involved the assessment of body composition for all study participants. Subsequently, the test battery was organized, and the time required for data collection was calculated. Table 2 presents the distribution of the tests conducted, based on the SBAFIT battery, following the methodology proposed by Mancha-Triguero, García-Rubio and Ibáñez [25].

A pilot test was conducted to familiarize participants with the evaluation protocols and to adjust the estimated data collection times. Subsequently, data collection was carried out over two separate sessions at the end of the season. Prior to the administration of the various tests, all participants were equipped with the necessary devices, and a standardized warm-up was implemented to ensure that players could perform at their maximum capacity during the assessments. This warm-up should never exceed 20 min. During this period, the necessary responses should be elicited so that athletes can face any challenging situation in the best possible condition. To achieve this, the athlete should complete 10 min of moderate activity, followed by 5 min of various dynamic stretching exercises and 3 min of active recovery or light activity [31].

All tests were conducted in the afternoon during the team’s regular training schedule, ensuring ecological validity of the assessment context. Players arrived at the facility after their habitual lunch, consumed at least three hours before testing; therefore, evaluations were not performed under fasting conditions. The assessments were carried out on the official indoor basketball court of the club, under stable environmental conditions. A minimum resting period of 15 min was provided between consecutive tests to allow recovery and to organize the subsequent protocol. During these intervals, athletes remained seated or engaged in light mobility exercises under staff supervision. This standardized procedure ensured that fatigue did not confound performance outcomes across the different assessments.

Finally, the processing and downloading of session data were performed using the SPro software (WIMU Pro). After processing the information, the relevant variables for this study were selected and extracted to create a comprehensive database containing the data of all participants. Subsequently, the variables were normalized according to the total duration of the task, enabling comparisons across participants independently of their individual performance times.

### 2.7. Statistical Analysis

A descriptive analysis of the selected specific variables was conducted. To visually represent the differences in physical performance between the U’14 and U’16 groups, radar charts were created for each test category. Each chart was constructed using the mean values of each variable per team, previously normalized to facilitate relative comparison between dimensions with different measurement units [32]. Normalization involved scaling each mean relative to the maximum value between the two groups, thus allowing for joint visualization on a common scale. An asterisk (*) was added next to the labels of variables where statistically significant differences were identified through hypothesis testing.

In order to determine the appropriate statistical model for hypothesis testing to compare the physical performance between the U’14 and U’16 teams, a normality test was performed on all quantitative variables from the physical test battery. Normality assumptions were verified using the ShapiroWilk test [33], which revealed that the variable Total Distance in the Aerobic Test did not follow a normal distribution. Consequently, this variable was analyzed using the non-parametric Mann–Whitney U test. Homogeneity of variances was further examined with Levene’s test. All variables satisfied this assumption except Total Distance in the Aerobic Test (*p* < 0.05). Therefore, while this variable was evaluated through non-parametric procedures, the remaining variables were analyzed using parametric methods. For the remaining variables, the independent samples Student’s t-test was applied [34]. All inferential analyses were performed using two-tailed (bilateral) tests, as no directional hypotheses were assumed regarding group differences. Given the small cohort size and the impossibility of expanding the sample, no prospective sample size calculation was performed. Instead, a post hoc analysis of statistical power (1-β) and effect sizes (Cohen’s d) was conducted to assess the robustness of each comparison.

Effect sizes were calculated to complement the interpretation of mean differences. For parametric comparisons, Cohen’s d index was employed, and its magnitude was interpreted according to the conventional thresholds proposed by Cohen [35]: small (0.2), moderate (0.5), and large (0.8). These cutoffs were retained as they remain the most widely used in sport science research, facilitating comparison with previous studies. For non-parametric analyses conducted with the Mann–Whitney U test, effect sizes were estimated using the rank-biserial correlation (*r*), which is recommended for this type of comparison. Statistical significance was set at α = 0.05, and exact *p*-values are reported, highlighting values below 0.001.

To analyze the relationship between the variables from the physical test battery, Pearson’s correlation coefficient (r) was employed. Normality of each variable was assessed using the Shapiro–Wilk test (α = 0.05). When at least one variable in a pair violated the assumption of normality, Spearman’s rank correlation coefficient (ρ) was used; otherwise, Pearson’s correlation coefficient (r) was applied. This approach ensured that parametric correlations were not inappropriately used for non-normal distributions. Only the total distance variable from the aerobic test exhibited a non-normal distribution. A correlation matrix was constructed, calculating all possible variable pairs, and redundant values (duplicated or symmetric correlations) were eliminated, retaining only the lower half of the matrix in accordance with standard methodological recommendations [36]. For interpreting associations, the following conventional cutoffs were considered: r ≥ 0.1 or r ≤ −0.1 (very weak correlation), r ≥ 0.3 or r ≤ −0.3 (marginal correlation), r ≥ 0.5 or r ≤ −0.5 (moderate correlation), r ≥ 0.7 or r ≤ −0.7 (strong correlation), and r ≥ 0.9 or *r* ≤ −0.9 (near-perfect correlation) [37]. Furthermore, the most significant correlations were visually represented using heat maps to facilitate interpretation.

To complement the descriptive statistics, the coefficient of variation (CV%) was calculated for each variable within both categories. The CV, defined as the ratio between the standard deviation and the mean multiplied by 100, provides a standardized measure of intra-individual variability. This metric enabled a more accurate comparison of the dispersion of outcomes between groups, independent of the absolute magnitude of the variable, and offered additional insights into developmental heterogeneity.

In addition to traditional descriptive and inferential analyses, violin plots were generated for each outcome variable. These graphical representations combine boxplot information with kernel density estimation, providing a detailed visualization of the distribution of individual scores within each category. Violin plots allowed the identification of differences in data dispersion between groups, thereby reinforcing the statistical comparisons and enhancing the interpretability of intra-individual variability.

All statistical calculations and the generation of radar charts were performed using the Python programming language (version 3.11), employing the Pandas library for data manipulation and Matplotlib, along with NumPy for graphical visualization.

## 3. Results

### 3.1. Body Composition Results

A preliminary assessment of participants’ body composition was conducted. The results revealed statistically significant differences (*p* < 0.001), with higher values in the U’16 group for body weight (77.3 ± 14.29 kg vs. 57.2 ± 9.95 kg), fat-free mass (65.4 ± 7.38 kg vs. 49.7 ± 7.12 kg), muscle mass (62.1 ± 7.04 kg vs. 47.2 ± 6.81 kg), and total body water content (48.25 ± 4.76 kg vs. 38.08 ± 4.33 kg). No significant differences were found in fat mass between the groups (*p* = 0.1895) (11.93 ± 8.18 kg vs. 7.55 ± 3.66 kg). All subjects analyzed were evaluated under the same conditions (environmental, day of the week, fasting) so that contamination/comparison between groups would not cause an error. This finding was interpreted within the context of an overall increase in body volume. The internal variability of the data (standard deviation) was moderate in both groups, although slightly higher in the U’16 cohort across all indicators, possibly reflecting greater individual differences characteristic of this stage of pubertal development.

Although the absolute values are detailed in the text, Figure 1 provides an immediate graphical representation of the trends across categories, facilitating a quick comparison of body composition profiles for each group by readers and practitioners.

The body composition analysis revealed substantial differences between the U’14 and U’16 players, particularly in body weight, fat-free mass, muscle mass, and total body water, all of which were significantly higher in the older group.

### 3.2. Descriptive Results

To provide a detailed overview of the distribution of performance across the two categories, violin plots were generated for each outcome variable (Figure 2). Violin plots were selected as they illustrate not only central tendency but also the full distribution and variability of the data, thereby providing a more comprehensive representation than traditional summary statistics. This approach allows a more precise visualization of both central tendency and intra-individual variability, facilitating the comparison between U’14 and U’16 players. U’14 players showed greater variability in aerobic (efficiency, total distance) and anaerobic (efficiency, total distance) measures, whereas U’16 players exhibited more homogeneous profiles. In jumping tests, U’16 players achieved higher mean values, with similar variability in the Abalakov test and reduced variability in the Multijump test, suggesting greater fatigue tolerance with age. Speed and agility outcomes revealed marked dispersion in U’14 players, particularly in sprint times with ball and agility measures, while U’16 players demonstrated tighter distributions consistent with improved coordination and stability. Finally, centripetal force results indicated greater heterogeneity in U’14 players and superior, more consistent performance in U’16 players, highlighting maturation-related gains in movement efficiency.

In addition to mean and standard deviation values, the coefficient of variation (CV) was calculated for each physical fitness outcome to better represent intra-individual variability. Overall, U’14 players exhibited higher CV values compared with the U’16 group, particularly in agility-related variables (CV = 6.8% vs. 4.4%) and sprint performance (8.0% vs. 4.7%), suggesting greater heterogeneity in physical development at earlier stages. By contrast, CV values for jumping tests were relatively similar between groups, indicating more consistent performance in explosive power across categories.

To further examine intra-individual variability, the coefficient of variation (CV%) was calculated for each performance variable across age categories (Table 3). Overall, U’14 players exhibited consistently higher CV values compared with U’16 players, particularly in agility and sprint-related measures, indicating greater heterogeneity in performance capacities at earlier developmental stages. In contrast, variables associated with lower-limb explosive strength (e.g., jumping tests) and aerobic capacity displayed relatively lower CVs in both groups, suggesting more homogeneous responses in these domains. These findings highlight the progressive reduction in variability with increasing age and maturation, reflecting the consolidation of physical performance profiles during adolescence.

### 3.3. Inferential Results of Differences

Table 4 presents the results of the inferential analysis conducted to compare differences between the U’14 and U’16 groups across each physical test. These results are complemented by Figure 3, which visually displays the descriptive profiles of the teams along with the identification of differences between groups.

The inferential analysis revealed statistically significant differences (*p* < 0.05) between groups, highlighting that the U’16 group significantly outperformed the U’14 group in total distance covered during the aerobic test (*U* = 0.0, *p* < 0.001, *r* = −0.84), sprint times (both with and without the ball), and average speed in the centripetal force test. These results reflect a superior development of aerobic capacity, linear speed, and resistance to centripetal forces in the U’16 group. The effect sizes, ranging from moderate to large, confirm the practical relevance of the identified differences. Conversely, no significant differences were found in variables related to efficiency, jump impulse, or external load, suggesting that these capacities may depend less on chronological age and more on individual characteristics or the specificity of training. These findings support the need for differentiated training approaches according to age group.

Although some variables showed statistically significant differences, others yielded small to moderate effect sizes (*d* = 0.20–0.55) and statistical power below 0.80; therefore, the lack of significance in these cases should be interpreted with caution and confirmed in studies with larger samples.

To enhance the visualization of intra-individual variability, violin plots were generated for each outcome variable (Figure 2). These plots provide a detailed view of the full distribution of individual scores, rather than summarizing them only with central tendency and spread. The violin plots confirm the presence of greater dispersion in the U’14 cohort, especially in agility and aerobic test performance, while U’16 players displayed more compact distributions around higher mean values. These visualizations reinforce the statistical findings and provide additional evidence of the role of biological maturation in reducing variability and consolidating performance profiles with age.

### 3.4. Correlation Analysis

Figure 4 presents the results of the correlation matrix. The correlation analysis revealed 37 significant associations (22 positive and 15 negative) between variables, indicating functional relationships among the various physical capacities evaluated.

The combined use of r and ρ did not alter the overall pattern of associations initially observed. However, the magnitude of certain coefficients was slightly reduced when applying Spearman’s ρ to non-normally distributed variables, underscoring the importance of adhering to normality assumptions.

The external load (Player Load) of the SIG-AER aerobic test positively correlated with the efficiency of the aerobic test (*r* = 0.544) and the times recorded in the sprint test (both with and without the ball), as well as with the best time with the ball in the agility test. On the other hand, it showed a negative correlation with the maximum speed achieved in the sprint test (*r* = −0.603).

The external load of the SIG-ANA anaerobic test also showed positive correlations with the total distance covered in both the aerobic (*r* = 0.516) and anaerobic (*r* = 0.664) tests, as well as with the accelerations per minute in the agility test. Negative correlations for this variable were found with decelerations per minute in the agility test (*r* = −0.638) and the maximum height in the Abalakov jump test (*r* = −0.695).

The correlations between the variables from the jump tests primarily occur internally. The mean number of jumps correlates with the maximum jump in the multijump test (*r* = 0.708) and the maximum height in the Abalakov test (*r* = 0.608). Similarly, the maximum heights from these tests are related (*r* = 0.621).

One of the most noteworthy correlations was observed between the variables from the sprint test, specifically between the best times for sprints with and without the ball (*r* = 0.947), and negatively between maximum speed and the best time with the ball (*r* = −0.94) and without the ball (*r* = −0.996). This suggests that players who achieve higher speeds tend to record better (lower) times in sprint tests, validating the consistency of these metrics.

The variables from the agility test showed strong internal correlations. Notably, there was a high positive correlation between accelerations and decelerations per minute (*r* = 0.968), as well as between the best times recorded with and without the ball (*r* = 0.94). Additionally, relationships were identified between the best times from the agility test, both with and without the ball, and the best times from the sprint test, both with and without the ball.

In this same line, the best times from the centripetal force test with and without the ball were strongly related (r = 0.794). Likewise, the best time for this test, with and without the ball, negatively correlates with the total distance in the aerobic test (r = −0.673; r = −0.55), meaning that less time in this test corresponds to more distance covered in the aerobic test.

The integration of violin plots and CV values also facilitated the interpretation of the correlation matrix by highlighting those variables with higher intra-individual dispersion. For instance, variables such as sprint times and agility measures showed both strong correlations and high within-group variability in U’14 players, suggesting that these tests may be particularly sensitive to maturational differences.

## 4. Discussion

The main objective of this research was to assess and compare the PF of male basketball players in the U’14 and U’16 youth development categories through the specific SBAFIT test battery. This study is particularly relevant given the scarcity of scientific data specifically addressing young players in formative stages of athletic development. The main findings reveal significant differences between categories in key aspects such as the total distance covered in the aerobic test, sprint times with and without the ball, and average speed in centripetal force tests, with superior performances recorded by U’16 players. However, no significant differences were found in variables related to shooting efficiency, jump impulses, or overall training loads. These findings suggest that the observed differences are predominantly linked to the biological and physical development associated with older age, whereas certain technical and neuromuscular capacities may depend more on specific and individualized training.

The differences found in the body composition of the two groups of players underscore the progressive influence of maturational development on physiological parameters that directly impact athletic performance. Greater muscle and lean mass in U’16 players may partially explain their superior results in aerobic capacity, speed, and strength-related tests, whereas the relatively lighter body mass of U’14 players could facilitate advantages in agility-based tasks. From a practical standpoint, these results highlight the necessity of tailoring training and conditioning programs to the maturational stage of players, ensuring that increases in body mass are accompanied by targeted interventions to preserve agility and coordination. Moreover, monitoring body composition across developmental stages provides valuable information for preventing imbalances that could limit performance or increase injury risk [38].

The descriptive results of this study reveal that U’16 basketball players exhibit higher performances in aerobic, anaerobic, and strength-related variables, specifically in total distance covered, maximum speed, and average speed in centripetal force tests, compared to U’14 players. Nevertheless, U’14 players demonstrated competitive outcomes in variables associated with explosive power, such as maximum height and impulse in jump tests. This performance increase with age has been extensively documented and is attributed not only to hormonal development but also to improvements in neuromuscular efficiency and cumulative motor learning [39,40]. These findings are consistent with previous research that identifies enhancements in physical capacities associated with biological maturation in adolescent athletes [7,8]. A similar pattern was observed by Guimaraes, Ramos, Janeira, Baxter-Jones, and Maia [13], who concluded that while early-maturing players tend to perform better in physical tests such as sprinting and agility, technical skills, such as passing and ball control, rely more heavily on accumulated training experience than on biological development. It is important to emphasize that some of the non-significant comparisons were accompanied by small effect sizes and low statistical power, which limits the robustness of the conclusions. These findings should be regarded as preliminary until they are corroborated with larger sample sizes.

Similarly, previous research indicates that the heterogeneity observed in younger categories reflects individual differences in the pace of physical and neuromuscular development [2]. These results are consistent with the findings of Gryko et al. [41], who observed that young female players closer to their Peak Height Velocity (PHV) exhibited significantly superior performance in specific tests such as 10 and 20 m sprints, agility, and jump tests, reinforcing the influence of biological development on motor capacities during formative stages. These findings highlight the importance of tailoring training programs according to age category and maturational status, emphasizing the development of specific physical capacities in accordance with each player’s evolutionary stage. In this context, coordination development should also be considered, as general coordination strongly relates to basketball-specific performance in youth players [15].

Biological or maturational age, beyond chronological age, plays a decisive role in physical performance during adolescence. In the present study, U’16 players, presumably undergoing more advanced pubertal development, demonstrated superior performance in the majority of physical tests, aligning with previous studies that have shown that earlier somatic maturation is associated with greater strength, power, and endurance performance [10,40,42]. Early maturation facilitates greater hormonal synthesis (particularly testosterone), increased muscle mass, and enhanced neuromuscular efficiency, thereby promoting superior physical performance [2,7]. The findings of this study are consistent with those reported in the meta-analysis by Albaladejo-Saura, Vaquero-Cristobal, Gonzalez-Galvez and Esparza-Ros [14], who concluded that athletes with advanced maturation achieve better results in strength and power tests, largely due to a more favorable hormonal environment and increased muscle mass. This natural advantage could bias talent identification and selection processes if the maturational status is not properly accounted for. Therefore, interpreting physical performance outcomes from a maturational, rather than purely chronological, perspective is essential for a fair and accurate assessment of players’ potential during formative stages.

The analysis of coefficients of variation provided additional insights into intra-individual variability across the different physical fitness domains. Our results showed that U’14 players consistently presented higher variability than U’16 players, particularly in agility and sprint-related outcomes, while jumping and aerobic measures displayed relatively lower dispersion in both groups. These findings are consistent with previous research indicating that variability in performance outcomes tends to decrease with age and biological maturation, reflecting the progressive stabilization of motor patterns and physical capacities during adolescence [2,17]. Monitoring intra-individual variability is crucial for practitioners, as it enables the identification of athletes who deviate from typical developmental trajectories, supports individualized training prescriptions, and helps to detect early signals of maladaptation or increased injury risk. Consequently, incorporating variability analyses into longitudinal monitoring frameworks may strengthen the ecological validity of player profiling and provide coaches with actionable information to optimize both training and competition strategies.

Inferential analysis revealed significant differences between the U’14 and U’16 teams, highlighting the superior performance of the U’16 group in key variables such as total distance covered in aerobic tests, sprint times (with and without the ball), and average speed in centripetal force tests. These results are in line with previous research that demonstrated improvements in physical capacities such as aerobic endurance, linear speed, and resistance to centripetal forces with increasing age and maturation among young players [27,29]. Furthermore, the absence of significant differences in shooting efficiency, jump impulse, and external load suggests that these variables may be more influenced by individual factors and specific training adaptations rather than chronological age. Regular evaluations are therefore necessary to adjust training loads and strategies to the specific developmental characteristics of each formative category.

The differences observed in body composition may partially explain some of the results found in this study. Specifically, the U’14 team exhibited better performance in the agility test compared to the U’16 team, which may be attributed to differences in body mass and muscle composition. Previous studies suggest that a lower body mass, characteristic of younger and less physically developed players, favors performance in agility tests due to greater ease in executing rapid movements and CODS [6,10]. Conversely, the increase in muscle mass and fat-free mass observed in U’16 players, although beneficial for strength and endurance, could to some extent limit agility due to the higher energetic and biomechanical cost associated with executing explosive movements. These findings highlight the need to emphasize specific exercises aimed at enhancing agility in players with higher levels of muscle mass and overall body mass.

A more favorable body composition, characterized by increased muscle mass and a lower proportion of fat mass, resulting from the biological development typical of adolescence, directly contributes to enhanced performance in physical tests such as speed, centripetal strength, and aerobic endurance. This body profile optimizes biomechanical and metabolic efficiency, allowing for greater force production, improved tolerance to prolonged exertion, and more effective recovery, ultimately translating into superior performance across most basketball-specific physical tests [3,4].

The correlation analysis conducted in the present study revealed significant associations among various SBAFIT battery variables, underscoring the existence of interdependent patterns among specific physical capacities. Particularly noteworthy were the strong correlations observed between best sprint times with and without the ball (*r* = 0.947) and agility test times (*r* = 0.940), as well as the negative correlation with maximum speed (*r* = −0.996), indicating that players who reach higher speeds tend to achieve better execution times in specific tests. Similarly, accelerations and decelerations per minute in the agility test showed a very high correlation (*r* = 0.968), suggesting that both variables reflect a shared pattern of neuromuscular control. These relationships were also evident in the jumping tests (Abalakov and Multijumps), where jump height and average jump height were strongly correlated (*r* > 0.600), indicating that these tests assess similar physical capacities. In practical terms, these findings suggest that certain dimensions of physical performance can be grouped into functional clusters, which is useful for a more integrated and specific assessment of players.

Although regression models could provide more detailed insights into the predictive contribution of each variable, the small sample size of this study limited their applicability and statistical robustness. Given the exploration scope and the primary aim of identifying functional interrelationships among test outcomes, correlations were considered the most appropriate analytical approach. Future studies with larger and more diverse samples are encouraged to apply regression analyses to strengthen predictive interpretations.

Several researchers have already recommended optimizing the physical evaluation process in young athletes by promoting more efficient batteries focused on the most impactful factors, thereby avoiding redundancy across tests [18,28]. Based on the observed relationships among variables from different tests, it is pertinent to propose a simplification of the test battery employed, as some variables provide redundant information. The high correlation between speed and agility results, both with and without the ball, suggests that it would be possible to select a single representative test to evaluate this capacity without compromising the validity of the assessment. This empirically based grouping of variables could contribute to the design of efficient minimal protocols, aligned with principles of resource economy and practical applicability in real training contexts [43,44].

In cases where high correlations are identified between tests measuring similar capacities, such as speed, agility, and centripetal force tests, it is recommended to prioritize those versions that incorporate basketball-specific technical gestures, such as dribbling. These tests offer greater ecological validity by more accurately replicating the technical-conditional demands of actual gameplay [25,26]. Including the ball not only increases specificity but also allows for the assessment of motor efficiency and technical control under high-intensity conditions, both of which are key aspects of competitive performance [4,38]. As pointed out by Sugiyama, Maeo, Kurihara, Kanehisa, and Isaka [24], there is no single test capable of covering all the motor demands of basketball; however, some tests, such as the T-Test or the V-Cut, can provide relevant information across multiple capacities if appropriately selected. Therefore, selecting tests that integrate basketball-specific movements, such as lateral shuffles, pivots, or diagonal CODS, can optimize the evaluation process by maintaining specificity while reducing redundancy.

Similarly, the relationship observed among the jumping tests supports the consideration of selecting a single test that reliably captures explosive jumping ability. This simplification would not only reduce the time required for assessment but also facilitate the planning and practical application of testing protocols in formative contexts, without compromising diagnostic quality.

Although partial correlations could provide additional insights by adjusting for players’ maturational stage, the limited sample size of this study restricts the statistical robustness of such analyses. Controlling for categorical covariates in small samples can considerably reduce statistical power and generate unstable estimates. For this reason, we opted to report bivariate correlations, which transparently illustrate the interrelationships among physical fitness variables while maintaining adequate interpretability. Future studies with larger and more heterogeneous samples are encouraged to apply partial correlation approaches to disentangle the effects of age category from the intrinsic associations between performance outcomes.

Potential sources of bias were minimized by including all available players to avoid selection bias, standardizing testing conditions (same facility, time, and evaluators), and employing validated instruments to ensure measurement reliability. Consequently, the findings should be considered exploratory, and statistical inferences are strictly confined to the sample analyzed, without any intent of population-level generalization. A limitation of this study is that no baseline equivalence analysis was performed, as group allocation was determined by age category; the primary aim was to compare developmental stages rather than to establish initial homogeneity. One of the main limitations of the present study lies in the small sample size, composed exclusively of players from a single club, which limits the generalizability of the findings to other competitive contexts or populations. Furthermore, prior experience in specific training was not precisely controlled, a factor that could significantly influence the observed performance. Finally, given the cross-sectional nature of the study design, it was not possible to analyze the progression of physical capacities over time or the impact of training programs on individual development.

For future research, it is suggested that the sample be expanded to include players from different clubs and regions, thereby incorporating a greater diversity of competitive contexts that would allow for more representative and generalizable comparisons. Additionally, it would be advisable to include objective indicators of biological maturation status and to document each player’s specific training history in order to deepen the analysis of interindividual differences. Finally, the implementation of longitudinal studies is recommended to evaluate the evolution of physical capacities over a season or a full formative cycle, which would facilitate the design of training programs tailored to the physical and maturational developmental trajectories of young players.

### Practical Applications

The findings of this research can be transferred to the practical and professional context of basketball. Coaches and strength and conditioning professionals should utilize validated test batteries and assessments to evaluate the PF of their players, enabling reproducibility and facilitating the comparison of results with other teams and researchers. The evaluation of PF in basketball should be as sport-specific as possible, avoiding the use of generic tests. Therefore, it is recommended to prioritize physical tests that incorporate ball handling (agility, speed, centripetal force), as they demonstrate greater sport-specific validity for youth basketball.

Optimizing evaluation time should be achieved by employing specific tests with higher informational value, thereby saving time in physical assessments without compromising diagnostic quality and facilitating their application in clubs and academies. Continuous monitoring of PF development is essential, both throughout a single season and longitudinally during the players’ growth process, taking into account their biological maturation. It is necessary to establish reference patterns for the physical performance of each age group in order to tailor training loads and objectives to the players’ developmental stages. Moreover, the use of specific test batteries could enhance talent identification processes, based on objective data.

## 5. Conclusions

This study provides an integrative and applied perspective on the physical performance of youth basketball players, highlighting the utility of the SBAFIT battery as a specific tool for the comprehensive evaluation of conditional profiles. This research is novel due to its comparative approach between developmental categories, a topic scarcely explored to date, as well as its correlational analysis between variables, which offers a foundation for optimizing evaluation protocols for basketball players. Older players (U’16) demonstrated superior performance in aerobic, anaerobic, and strength capacities, while U’14 players exhibited competitive values in agility and jumping tests. Differences were primarily associated with biological maturation status and the development of muscle mass resulting from the natural growth process. The players from the analyzed teams differed significantly in key physical performance variables, such as total distance covered, average speed, and displacement times, due to the logical progression of physical development across age groups. Strong correlations were identified among multiple physical test variables. Consequently, it is recommended to optimize evaluation processes by eliminating tests that assess similar factors, thereby maintaining diagnostic validity and promoting a more efficient assessment of physical performance in young basketball players. It is advised to employ a single test to evaluate jumping capacity and to prioritize tests involving ball handling in assessments of speed, agility, and centripetal force, as these offer greater specificity.

## Figures and Tables

**Figure 1 jfmk-10-00382-f001:**
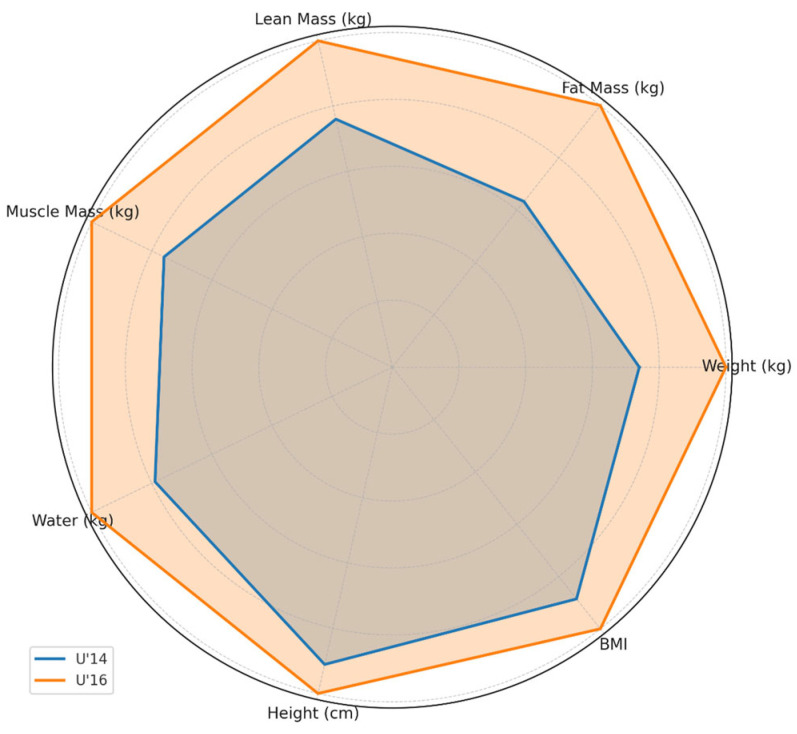
Body composition of the participating teams (comparative visualization; full numerical values are provided in the text). Note: Variables include body weight (kg), fat mass (kg), fat-free mass (kg), muscle mass (kg), total body water (kg), and Body Mass Index (BMI, kg/m^2^).

**Figure 2 jfmk-10-00382-f002:**
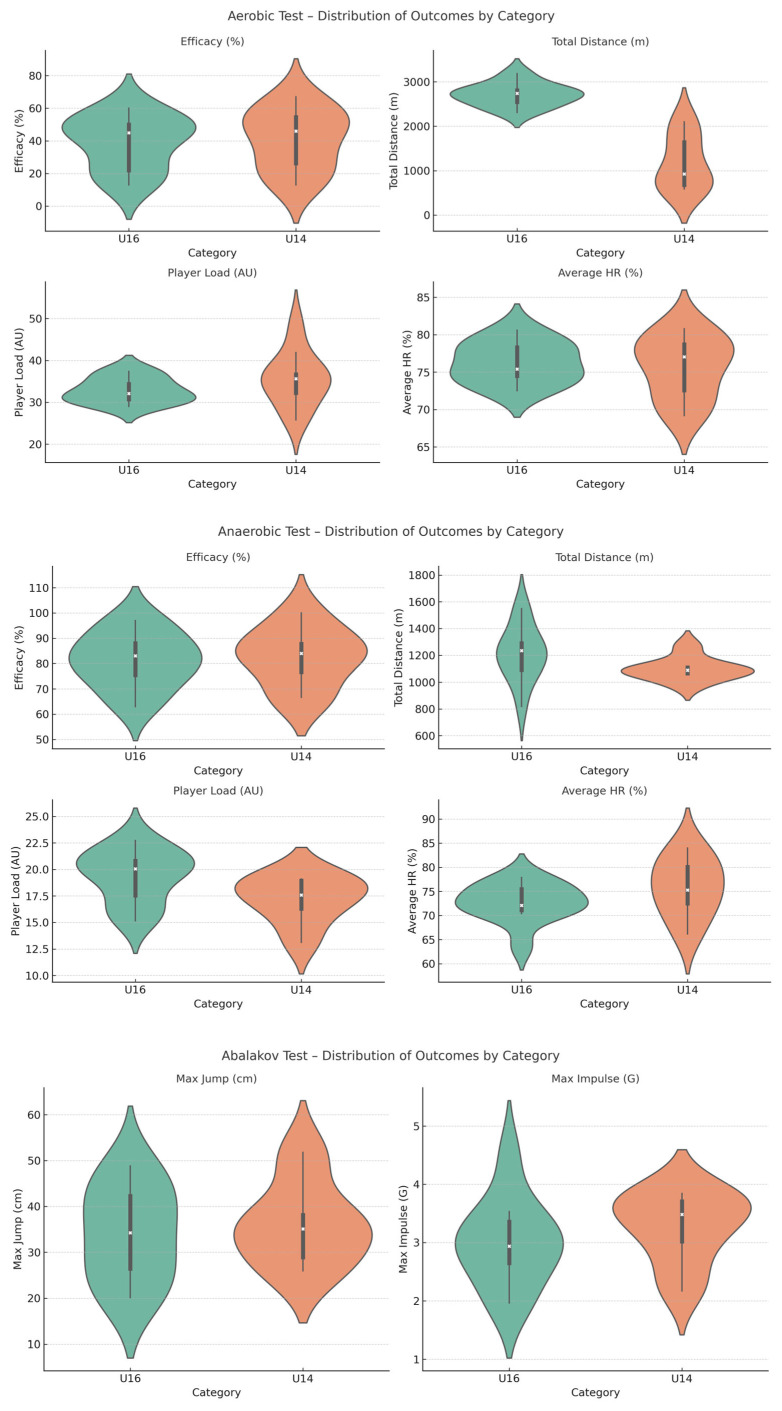
Violin plots displaying the distribution of physical fitness outcomes by category (U’14 vs. U’16).

**Figure 3 jfmk-10-00382-f003:**
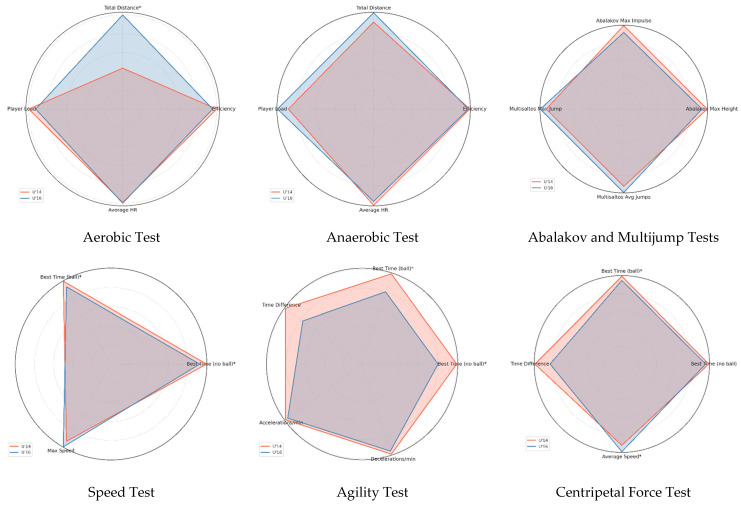
Comparative profiles of teams by SBAFIT test battery.

**Figure 4 jfmk-10-00382-f004:**
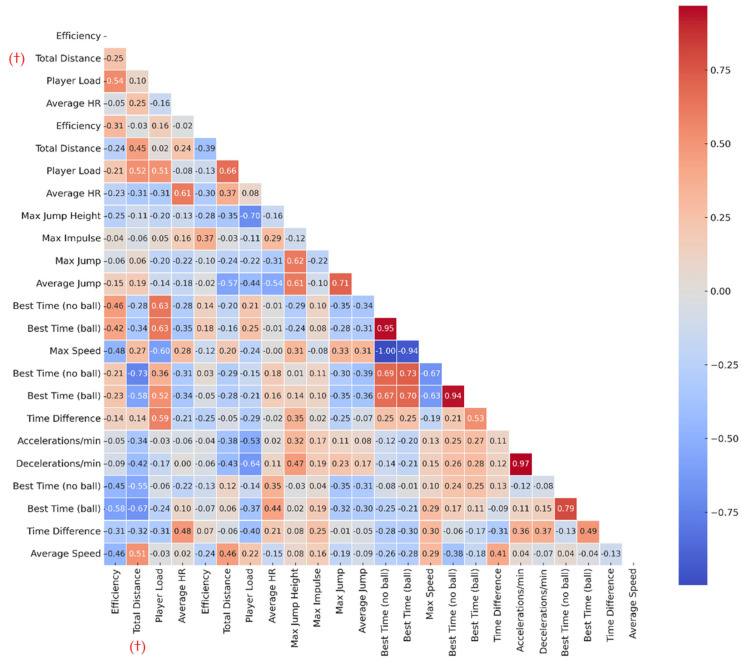
Correlation matrix of the variables from the SBAFIT test battery. Note: Correlations between performance variables (r = Pearson’s correlation for normally distributed pairs; ρ = Spearman’s rank correlation for non-normally distributed pairs). Coefficients marked with (†) in the Total Distance row and column of the aerobic test correspond to ρ.

**Table 1 jfmk-10-00382-t001:** Definition of the study variables.

Test	Variables	Units	Description
Aerobic test	Efficiency	Percentage	Successful throws.
	Total distance	Metres	Total distance covered.
	Player load	Count	Player workload.
	Heart rate average	Percentage	Personal average heart rate
Anaerobic test	Efficiency	Percentage	Successful throws.
	Total distance	Metres	Total distance covered.
	Player load	Count	Player workload.
	Heart rate average	Percentage	Personal average heart rate
Abalakov test	Maximum jump	cm	Maximum height reached.
	Maximum impulse	G Force	Maximum force applied in the jump
Multijump test	Maximum jump	cm	Maximum height reached.
	Average jump height	cm	Average height of jumps.
Velocity test	Best time without the ball	Seconds	Shortest time recorded without the ball.
	Best time with the ball	Seconds	Shortest time recorded with the ball.
	Maximum speed	km/h	Maximum speed of the test
Agility test	Best time without the ball	Seconds	Shortest time recorded without the ball
	Best time with the ball	Seconds	Shortest time recorded with the ball.
	Difference	Seconds	Difference between best time with the ball and best time without the ball.
	Acelerations/minute	Count	Average accelerations per minute.
	Decelerations/minute	Count	Average decelerations per minute.
Centripetal force test	Best time without the ball	Seconds	Shortest time recorded without the ball
	Best time with the ball	Seconds	Shortest time recorded with the ball.
	Difference	Seconds	Difference between best time with the ball and best time without the ball.
	Average Speed	km/h	Average speed of the test.

**Table 2 jfmk-10-00382-t002:** Distribution of the SBAFIT test battery.

**Test Battery**			
Day 1	Lower-Lomb Strength Test (Abalakov y Multisaltos)	Arc Test	Aerobic Test
Day 2	Sprint Test	Agility Test	Anaerobic Test

**Table 3 jfmk-10-00382-t003:** Coefficient of variation (CV%) for physical fitness variables by category (U’14 vs. U’16).

Test	Variables	U’14 (CV%)	U’16 (CV%)
Aerobic test	Efficiency	45.4	41.8
	Total distance	51.7	9.6
	Player load	18.3	9.2
	Heart rate average	5.4	3.6
Anaerobic test	Efficiency	14.0	13.4
	Total distance	7.8	17.0
	Player load	12.8	12.7
	Heart rate average	8.1	5.4
Abalakov test	Maximum jump	24.7	30.6
	Maximum impulse	18.3	24.9
Multijump test	Maximum jump	21.2	20.7
	Average jump height	11.0	16.2
Velocity test	Best time without the ball	8.0	4.7
	Best time with the ball	6.8	4.9
	Maximum speed	8.4	4.9
Agility test	Best time without the ball	6.8	4.4
	Best time with the ball	9.2	6.8
	Difference	101.6	111.7
	Accelerations/minute	4.3	5.2
	Decelerations/minute	4.4	4.6
Centripetal force test	Best time without the ball	5.1	4.2
	Best time with the ball	4.9	4.4
	Difference	42.5	39.5
	Average speed	8.5	5.3

Note. Values represent the coefficient of variation (CV%), calculated as (SD/Mean × 100). Higher CV values indicate greater intra-individual variability within each group.

**Table 4 jfmk-10-00382-t004:** Results of the inferential analysis to identify differences between teams.

Test	Variable	Contrast Model	Statistic	*p*	Cohen’s d/ Rank-Biserial Correlation (r)	Power
Aerobic test	Efficiency	t de Student	0.225	0.824	0.103	0.055
	Total distance	Mann–Whitney U	<0.001	<0.001	−3.67	−0.84
	Player load	t de Student	1.168	0.263	0.518	0.187
	Heart rate average	t de Student	−0.409	0.687	−0.184	0.067
Anaerobic test	Efficiency	t de Student	0.230	0.821	0.108	0.056
	Total distance	t de Student	−1.669	0.116	−0.684	0.265
	Player load	t de Student	−1.96	0.069	−0.925	0.436
	Heart rate average	t de Student	1.198	0.260	0.639	0.238
Abalakov test	Maximum jump	t de Student	0.523	0.607	0.234	0.079
	Maximum impulse	t de Student	0.895	0.382	0.401	0.136
Multijumps test	Maximum jump	t de Student	−0.793	0.437	−0.355	0.117
	Average jump height	t de Student	−1.241	0.233	−0.555	0.217
Speed test	Best time without the ball	t de Student	2.272	0.046	1.164	0.675
	Best time with the ball	t de Student	2.489	0.029	1.234	0.725
	Maximum speed	t de Student	−2.150	0.055	−1.08	0.610
Agility test	Best time without the ball	t de Student	7.549	<0.001	3.896	1.000
	Best time with the ball	t de Student	5.509	<0.001	2.784	1.000
	Difference	t de Student	0.515	0.614	0.243	0.080
	Accelerations/minute	t de Student	1.311	0.210	0.598	0.220
	Decelerations/minute	t de Student	1.613	0.131	0.766	0.330
Centripetal force test	Best time without the ball	t de Student	1.726	0.102	0.772	0.373
	Best time with the ball	t de Student	2.291	0.034	1.025	0.583
	Difference	t de Student	1.030	0.317	0.461	0.164
	Average Speed	t de Student	−2.412	0.028	−1.079	0.626

## Data Availability

Data are unavailable due to privacy or ethical restrictions.

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
