# Peer review of "Physical Fitness Profiling of Youth Basketball Players by Developmental Stage: A Case Study"

_jfmk, 2025, doi:10.3390/jfmk10040382_

Round 1

Reviewer 1 Report

Comments and Suggestions for Authors

Dear authors, I carefully reviewed the original work entitled: “Physical Fitness Profiling of Youth Basketball Players by 2 Developmental Stage: A Case Study”; where you compared physical fitness among U´14 and U´16 basketball players, assessing their interrelationship.  The reported findings are quite interesting and novel as a scarce number of studies have focused on the assessed populations as you well mentioned in the introduction. Moreover, the data provides a basis for future large-cohort and longitudinal investigation that could explore performance development across distinct maturity stages in male basketball players. Detailed comments and recommendations are provided below, emphasizing the need of further transparency among methods applied to assess performance variables, statistical analysis to explore interrelationship among performance metrics, and physiological basis for higher aerobic performance and speed in U´16 players.

Conceptualization of the study

Introduction

  • Line 39— No movement is exclusively sustained through aerobic or anaerobic pathways. Please rephrase to avoid interpretation bias.

Methods

Weight, body composition, and anthropometrics

  • Please specify whether body composition was assessed under fasting conditions. Also, specify the validity and accuracy of the TANITA analyzer for assessing body composition, indicating if such device uses tetrapolar and multifrequency method. If so, it would be great if you could provide player´s differences in segmental body fat and lean mass distribution which would be of particular interest to better understand players’ performance.
  • Avoid presenting any body composition comparisons in methods. Instead, provide such findings in the results and discuss its implications.

SBAFIT Battery test

  • Most of your readers won’t be familiarized with the SBAFIT battery test. Thus, provide either a graphical or textual description of the tests, beyond mere description of the retrieved outcomes and their corresponding units. This will ensure transparency and repeatability of your applied methods. Perhaps you might provide this as a supplementary file, but I recommend you include it in the main text to facilitate the readability of your data.
  • Describe the cool down or resting period between the performed tests.
  • Also specify whether the tests were performed under fasting conditions, the exact timing (morning or evening), and location of the evaluations.

Statistical analysis and results

Intra-individual variability of players performance

  • Considering the relevance of intra-individual variation, violin plots or raincloud plots should be used instead of boxplots to represent performance variabilities among players categories.
  • Coefficient of variations or outcomes variance should be provided to represent intra-individual variation, instead of simply referring to visual analysis of standard deviations

For better visualization of such variability, creating one panel per outcome would be more convenient than adding all assessed variables in a single panel.

Performance outcomes interrelationship

  • Partial correlation (adjusting by players age or category) would be recommended in this case given that significant differences were observed in performance outcomes between groups. Only if the variables correlation is not influenced by players category, the data can be pooled together, preventing interpretation bias.

Discussion

Differences in aerobic capacity

  • Specify those physiological mechanisms explaining a higher aerobic capacity and speed in U´16 players, beyond discussing mere differences in body composition such as muscle mass.

Limitations and future directions

  • Lack of female players inclusion is a key limitation of the present study. Please address.
  • Principal component analysis in further large-cohort studies should be recommended to see how these performance variables group together, avoiding spurious correlations.

Figures

  • All figures have a poor resolution. Please improve its visual quality.
  • Add BMI units to figure 1.

Reviewer 2 Report

Comments and Suggestions for Authors Methodology   In general, the methodology is appropriate; however, I followed the STROBE checklist, which recommends including a specific section on potential sources of bias.   Regarding the statistical analysis, t-tests were used, but the authors did not verify the assumption of equal variances using Levene’s test.   The authors report only the use of means but do not include any measures of dispersion such as standard deviation, range, median, or confidence intervals. These are essential to understand the overall behavior of the data models.   Concerning effect size, there are two issues. First, the authors refer to the traditional interpretation thresholds of Cohen’s d, but more recent guidelines suggest more conservative cutoffs. Secondly, for the Mann–Whitney U test, Cohen’s d is not appropriate; in this case, Cramér’s V should be used instead.   Regarding the p-values, the cutoff threshold (e.g., 0.05 or 0.001) is not specified. Moreover, it is not stated whether the inferential tests were one-tailed or two-tailed (i.e., bilateral significance).     Results   In the baseline descriptive figures, only means are reported. Although a graphical standard deviation is possibly shown, it is unclear, as the statistical analysis section in the methodology does not provide details. Nevertheless, based on the authors' statements, they used Student’s t-tests and Mann–Whitney U tests along with effect size measures. Therefore, it is unclear why no inferential analysis was conducted to assess baseline heterogeneity. This is important to determine whether the two groups (U14 vs. U16) were equivalent and comparable at baseline.   As for inferential tests, the p-values are not reported correctly. For example, Table 3 lists "0.000", which should be written as "<0.001". Similarly, Cohen’s d is only valid for Student’s t-test. For variables analyzed with the Mann–Whitney U test, Cramér’s V is the appropriate measure.   Regarding Figure 4 and the correlation matrix, it would be more informative to conduct a regression model analysis. This approach would allow assessment of the strength of association and the contribution of each variable to the predictive model, rather than relying on correlations, which only indicate potential associations without causal inference.     References   The references are outdated; I counted 28 out of 44 published before 2020 (i.e., more than five years old). This raises concerns about the novelty and currency of the review, especially in a topic that has already been extensively studied and detailed in the literature. The reference list should be updated.

Reviewer 3 Report

Comments and Suggestions for Authors

The study is quite clear-cut, but the presentation and interpretation are somewhat on the lower quality side.  The construction of the manuscript is more or less ok, but the introduction is clearly too long and not focused on the issue enough; the discussion also lacks focus and critical evaluation of the results/findings, as well as a more transparent appreciation of the limitations. Conclusions are too narrative and give little impression about the study outcomes - please be more concise and specific, even if some practical applications and perspectives could be left if in short. Please address these issues properly.

Some more points on which the paper could be improved: 

  • Please do not rely that heavily on the Tanita data for body composition since these are clearly not precise. For instance, the estimates of body fat etc. rely heavily on the height and age typed there, thus comparison between the athletes' groups is more or less meaningless in your case. You can use some of the fat mass markers, but presenting them as one of the major findings in the graphs is not recommended by me.
  • Some words in the tables are in Spanish - please translate properly. What is the "Anaerobic efficacy" there? Please use proper terms.
  • Please redo the figures as the text there is too small to be read conveniently. Also provide there the units of measurement where appropriate.
  • Accelerations and decelerations counts -- I wonder does that makes sense to present them both since I tend to think that they should have the same numbers for a period (in analogy with a flight phase and a contact phase during running...)
  • In the Methods, please make sure to present the details of the tests conducted, since, as it is now, it is not possible to comprehend properly, not to mention conducting the same tests in another location.
  • Are you sure you have touched the main papers in the field? I miss some, such as PMID: 24665800.

Round 2

Reviewer 2 Report

Comments and Suggestions for Authors

I congratulate the authors, they considerably improved their manuscript and clarified almost all the doubts, and the suggestions were corrected when necessary.

Reviewer 3 Report

Comments and Suggestions for Authors

You have indeed addressed some of my comments.
